# Advances in Intralesional Therapy for Locoregionally Advanced and Metastatic Melanoma: Five Years of Progress

**DOI:** 10.3390/cancers15051404

**Published:** 2023-02-23

**Authors:** Danielle K. DePalo, Jonathan S. Zager

**Affiliations:** 1Department of Cutaneous Oncology, Moffitt Cancer Center, Tampa, FL 33612, USA; 2Department of Oncologic Sciences, University of South Florida Morsani College of Medicine, Tampa, FL 33612, USA

**Keywords:** advanced melanoma, in transit metastases, intralesional therapy, intratumoral therapy, local therapy, locoregional therapy, melanoma, metastatic melanoma, regional therapy, regional chemotherapy

## Abstract

**Simple Summary:**

The prognosis for patients with locoregionally advanced and metastatic melanoma remains poor despite advances in systemic therapy. By delivering therapeutics directly to the site(s) of disease, intralesional therapies have the advantage of delivering the oncolytic agent directly into the metastatic melanoma while minimizing systemic side effects and resistance. Within the past 5 years, numerous potential intratumoral therapies have been investigated, though few have reached phase 2 clinical trials. We present a discussion of the scientific rationale for and status of intralesional therapies that have reached phase 2 or later clinical trials within the past 5 years in order to inform providers about current and upcoming intralesional therapeutic options for advanced melanoma.

**Abstract:**

Locoregionally advanced and metastatic melanoma are complex diagnoses with a variety of available treatment options. Intralesional therapy for melanoma has been under investigation for decades; however, it has advanced precipitously in recent years. In 2015, the Food and Drug Administration (FDA) approved talimogene laherparepvec (T-VEC), the only FDA-approved intralesional therapy for advanced melanoma. There has been significant progress since that time with other oncolytic viruses, toll-like receptor agonists, cytokines, xanthene dyes, and immune checkpoint inhibitors all under investigation as intralesional agents. Further to this, there has been exploration of numerous combinations of intralesional therapies and systemic therapies as various lines of therapy. Several of these combinations have been abandoned due to their lack of efficacy or safety concerns. This manuscript presents the various types of intralesional therapies that have reached phase 2 or later clinical trials in the past 5 years, including their mechanism of action, therapeutic combinations under investigation, and published results. The intention is to provide an overview of the progress that has been made, discuss ongoing trials worth following, and share our opinions on opportunities for further advancement.

## 1. Introduction

While early-stage localized melanoma disease has a 5-year survival of greater than 90%, regional and distant disease have 5-year survival rates of 66% and 27%, respectively [1]. Therefore, there remains significant room for improvement in the treatment of advanced melanoma. Many of the major advancements in the treatment of advanced disease have been with systemic immune checkpoint inhibitors (ICIs) and targeted therapies [2,3,4]. These therapies have revolutionized the treatment of advanced disease, which previously had an even poorer prognosis; however, they can have permanent, life-altering side effects in some and responses are limited [2,3,4,5]. In patients with locoregionally advanced and metastatic melanoma that is unresectable but amenable to injection, intralesional therapy provides a promising alternative or adjunct to systemic therapy.

Intratumoral therapy has been explored for the treatment of melanoma since the 1970s. Early investigations included the injection of bacillus Calmette–Guerin (BCG), a live, attenuated strain of Mycobacterium bovis developed as a tuberculosis vaccine, as an intralesional therapy for melanoma in 1974 [6,7]. More recent randomized trials have not supported the efficacy of BCG [8], and it has fallen further out of favor due to significant systemic toxicity [7,9]. However, since that time, innumerable other intralesional therapies have been and continue to be investigated for the treatment of advanced melanoma. To limit the scope of this discussion to the most current and clinically relevant therapies, the focus of this review is on intralesional therapies that have reached phase 2 or later clinical trials within the past 5 years.

To identify clinical trials, the clinicaltrials.gov Advanced Search feature was used. The condition or disease was specified as melanoma and phase 2, phase 3, and phase 4 were selected. Key terms searched were: intralesional, intratumoral, inject, xanthene, PV-10, Rose Bengal, intratumoral interleukin, intratumoral interferon, Aldesleukin, tavokinogene telseplasmid, tavo, electroporation, Darleukin, L19IL2, Daromun, L19TNF, stimulator of interferon genes, STING, DMXAA, MIW815, MK1454,, oncolytic virus, talimogene laherparepvec, T-VEC, GM-CSF, vusolimogene oderparepvec, RP1, OrienX010, Oncos-102, BT-001, canerpaturev, C-REV, TBI-1401, HF-10, coxsackievirus A21, CA21, CAVATAK, V937, gebasaxturev, lerapolturev, PVSRIPO, Telomelysin, OBP-301, Voyager-V1, VV1, VSV-IFNβ-NIS, VSV-IFNβTYRP1, Ad-p53, RheoSwitch, Ad-RTS-hIL-12, IXN-2001, veledimex, IXN-1001, oca511, Toca FC toll-like receptor, TLR, tilsotolimod, IMO-2125, SD-101, vidutolimod, CMP-001, cavrotolimod, AST-008, Lefitolimod, MGN1703, NKTR-262, G100, CV8102, LHC165, intratumoral ipilimumab, LL37, MAGE-A3, Hiltonol, poly-ICLC, APX005M, ABBV-927, dendritic cell, INT230-6, and TTI-621 polidocanol. Trials were included if they had published results in 2018 or later or were ongoing as of December 2022. Trials were excluded if at least one agent was not delivered intralesionally.

## 2. Xanthene Dyes

### 2.1. PV-10

PV-10 is a formulation of 10% Rose Bengal xanthene dye in saline. The dye was first found to have antitumor effects when it was observed to have a dose-dependent survival benefit in a strain of mice with a propensity for spontaneous tumor development that were used to test its safety as a food dye [10]. PV-10 is theorized to cause photolytic release of lysosomal enzymes resulting in tumor cell lysis and has been found in vitro to cause cell lysis in melanoma cell lines, but not in normal fibroblasts [11,12]. Bystander lesion response has also been observed with intralesional PV-10 injection and is theorized to result from High Mobility Group Box 1 activation and the maturation of dendritic cells and tumor-specific CD-8+ T cells [13,14,15,16]. 

In 2015, a phase 2 clinical trial of single-agent intratumoral PV-10 in 80 patients with treatment-refractory American Joint Committee of Cancer 8th Edition [17] stage III-IV melanoma found an overall response rate (ORR) of 51% with an ORR of 33% in uninjected lesions [18]. AEs were observed in all patients, mostly grade 1 to 2, while 15% experienced possibly related grade 3 adverse event (AE); the most common AEs were injection site pain (80%), edema (41%), and vesicles (39%). This was continued in two single-institution studies that reported ORRs of 53–87% with a 50% response in non-injected lesions and consistent AE profiles [19,20]. The dosing schedule examined, intralesional injections on Day 0 then repeated on Weeks 8, 12, and 16 for additional or incomplete responding lesions, makes this therapy particularly appealing. The addition of external beam radiotherapy (XRT) to intralesional PV-10 was examined in stage IIB-IV melanoma in a phase 2 trial at a single institution [21]. A total of 98 target lesions in 15 patients were treated, the ORR was 87%, and all patients experienced at least one grade 1 or 2 AE, including injection site pain (87%), swelling (60%), and blistering (20%) with one patient experiencing a grade 3 AE of injection site pain.

A phase 3, randomized controlled trial (NCT02288897) sought to compare PV-10 to investigator choice chemotherapy (dacarbazine or temozolomide) or T-VEC, but was terminated early due to an inadequate rate of enrollment and the changing landscape of therapy for advanced melanoma.

Since PV-10 exerts its antitumor effects through T cell activation [13], the addition of immune checkpoint inhibitors provides a potential pathway for overcoming resistance or potentiating PV-10′s effects. The efficacy of intralesional PV-10 with systemic pembrolizumab, anti-programmed cell death protein 1 (PD-1) antibody, in stage IIIB-IVM1c unresectable melanoma is undergoing evaluation in a phase 1b/2 trial (NCT02557321) in which intralesional PV-10 is dosed every 3 weeks with intravenous pembrolizumab for up to 5 weeks followed by intravenous pembrolizumab alone (Table 1). In phase 1b, 21 ICI-naive patients experienced a 67% ORR with primarily grade 1 and 2 injection-site-related AEs and one grade 3 AE related to PV-10 of injection site pain as well as AEs consistent with pembrolizumab’s known toxicity profile [22]. An expansion phase 1b cohort in 14 evaluable ICI-refractory patients had an ORR of 29% and AEs were consistent with known drug profiles, primarily grade 1 or 2 injection site events related to PV-10 and grade 1–3 immune-related events related to pembrolizumab [23]. Phase 2 with arm 1, PV-10 with pembrolizumab, and arm 2, pembrolizumab alone, is ongoing. 

### 2.2. Cytokines

#### 2.2.1. Interleukin-2 (IL-2, Aldesleukin)

IL-2 acts to activate numerous pathways modulating lymphocyte proliferation, activity, and survival by binding the IL-2 receptor [31]. It received Food and Drug Administration (FDA) approval in 1998 for the treatment of advanced melanoma; however, its systemic application has been limited by significant side effects at efficacious doses [32]. Intralesional injection of IL-2 is an appealing way to minimize systemic effects while locally delivering high doses of the drug. This premise was first investigated in 1994 [33], and in 2010 a phase 2 trial in 48 evaluable patients reported a complete response (CR) of 69%, though the response was not observed in any uninjected lesions [34]. Only grade 1/2 AEs were reported, including injection site reaction and pain in most patients as well as fever (58%), fatigue (36%), and nausea (34%).

The addition of ICIs to cytokine therapy should theoretically potentiate cytokine effects by antagonizing immune regulatory pathways [31]. In 2017, a phase 2 trial (NCT01480323) sought to investigate the combination of intratumoral IL-2 plus systemic ipilimumab, an anti-CTLA-4 antibody, in 15 patients with treatment-refractory stage IV melanoma, but found a disappointing ORR of 0%, a disease control rate (DCR) of 20%, and grade 3/4 AEs in 40%, most commonly fatigue and pain (excluding injection pain) [35] The combination of systemic ICI with intratumoral IL-2 is under further investigation in a phase 1/2 trial (NCT03474497) evaluating intralesional IL-2 in combination with systemic pembrolizumab and hypofractionated radiotherapy in patients with metastatic melanoma and other solid tumors who failed to respond or progressed on previous anti-PD-1/PD-L1 therapy (Table 1). One limitation to the use of IL-2 in the above-mentioned trials is the frequent dosing schedule of injections, two times per week. Every-2-week dosing is under investigation in a phase 2/3, randomized controlled trial (NCT03928275), a 2-stage study of intralesional IL-2 versus combination intralesional IL-2 and BCG in stage IIIC-IVM1a melanoma. 

#### 2.2.2. Tavokinogene Telseplasmid (Tavo)

Interleukin-12 (IL-12) bridges the innate and adaptive immune systems, stimulates Interferon-gamma (IFN-γ), regulates natural killer (NK) and T cell production, and promotes T helper type 1 (Th-1) differentiation [36,37]. Systemic administration of IL-12 has been found to have high toxicity with limited efficacy [37,38]. Tavokinogene telseplasmid (tavo) is a plasmid that encodes IL-12 that is delivered intratumorally and is typically combined with electroporation (EP) to improve plasmid cell uptake by increasing cell permeability [39,40]. Simply, this results in increased local IL-12 and IFN-γ expression, which activates the innate and adaptive immune systems [39,41,42]. 

Intratumoral tavo-EP alone in stage III-IV melanoma was investigated in a phase 2 trial (NCT01502293) (Figure 1). The results published in 2020 included a CR rate of 18% and ORR of 36% in the 28 patients in the standard dosing cohort, with a median PFS of 3.72 months [42]. The most common AEs reported were procedural pain in 80% and various injection site reactions. Since cytokines increase tumor-specific T cells, the addition of ICIs should potentiate their effects by inhibiting immune regulatory pathways [31]. This combination is under investigation with intratumoral tavo-EP and intravenous pembrolizumab in a phase 2 trial (NCT03132675) in 54 ICI-refractory patients with stage III/IV melanoma (Table 1). On interim analysis, ORR was 30% with a reduction in non-injected tumor burden seen in all 12 patients with non-injected disease and the most common AEs were low-grade fatigue, procedural pain, and diarrhea [24]. Though tavo-EP injection occurs on Days 1, 5, and 8 of each cycle, cycles span for 6 weeks, making overall dosing frequency more manageable. 

### 2.3. Antibody–Cytokine Fusion Proteins

#### 2.3.1. Darleukin (L19IL2)

As previously discussed, IL-2 acts to increase tumor immunogenicity but is not well tolerated when delivered systemically at efficacious doses [31,32]. A fusion protein of IL-2 to L19, a monoclonal antibody targeted to an angiogenesis marker, L19IL2, enables preferential delivery and activation of the cytokine within the tumor cells [43]. It has been investigated as a single agent in a phase 2 trial (NCT01253096) in 24 evaluable patients with unresectable stage IIIB/C melanoma and found to generate a CR in 25% at lower doses than non-targeted IL-2 with the most common toxicity being injection site reaction, seen in 76%, with few grade 3 cases [44]. Other AEs were seen in 25% or less and included fatigue, edema, and fever. The addition of intratumoral L19IL2 to systemic dacarbazine vs. systemic dacarbazine alone is now under investigation for the treatment of stage IVM1a-b melanoma in a phase 1/2 trial (NCT02076646) (Table 1). 

#### 2.3.2. Daromun (L19IL2 and L19TNF)

Daromun is the combination of the previously mentioned, L19IL2 and L19TNF, a fusion protein of L19 and human recombinant tumor necrosis factor-alpha (TNFα). The addition of L19TNF acts synergistically in murine models to enhance the antiangiogenic and, therefore, antitumor effects [45]. A phase 2 trial (NCT02076633) of intratumoral daromun injection weekly for 4 weeks in stage III-IVM1a melanoma in 2015 found that of 20 evaluable patients, 50% had PR, 25% had SD, and 20% had PD for an ORR of 55% and DCR of 80%. Additionally, in 13 lesions not amenable to injection, 54% demonstrated CR [46]. Injection site reaction was the most common AE, seen in 73%, and was the only grade 3 AE reported. Other common grade 1 and 2 AEs were fever, headache, edema, and erythema. 

The efficacy of intralesional daromun in the neoadjuvant setting with systemic investigator’s discretion ICI compared to neoadjuvant systemic ICI alone is now under investigation in a phase 3 trial (NCT03567889) in resectable stage IIIB/C melanoma in the United States and Europe (Table 1) [47]. A similar trial (NCT02938299) is ongoing at European centers with neoadjuvant intratumoral daromun versus surgery alone. 

### 2.4. Oncolytic Viral Therapies

#### 2.4.1. Talimogene Laherparepvec

Talimogene laherparepvec (T-VEC) is a live-attenuated herpes simplex virus type-1 (HSV1) that has been engineered to express the gene for granulocyte-macrophage colony-stimulating factor (GM-CSF) [48]. Two mutations in the engineered HSV1, *γ34.5* and *α47* gene deletions, enable the virus to selectively replicate in tumor cells and enhance the immune response, respectively [49]. T-VEC enters tumor cells through herpes virus glycoproteins and ultimately results in cell lysis. Lysis of injected cells releases viral particles, tumor-derived antigens, and GM-CSF, promoting both cell-mediated and humoral immune responses and linking innate and acquired immune systems through T cell proliferation and activation and DC growth and maturation [49,50]. While the results of intralesional injection of GM-CSF have been underwhelming [51,52], intralesional T-VEC has received FDA approval. This is based on the results of the OPTiM phase III trial in stage IIB-IV melanoma, in which 47% of injected lesions, 22% of uninjected non-visceral lesions, and 9% of uninjected visceral lesions had CR [53]. The most common adverse events seen in those treated with T-VEC were fatigue (50%), chills (49%), and pyrexia (43%), with an 11% incidence of grade 3 or 4 treatment-related AEs including cellulitis (2%), fatigue (2%), and vomiting (2%) [54].

Single-agent T-VEC neoadjuvant therapy with surgery (arm 1, *n* = 76) versus surgery alone (arm 2, *n* = 74) in resectable Stage IIIB-IVM1a melanoma was compared in a phase 2 randomized controlled trial, NCT02211131, which demonstrated that 2-year recurrence-free survival (RFS), the primary endpoint, was improved with the addition of neoadjuvant T-VEC, 29.5% vs. 16.5% (overall hazard ratio 0.75, 80% confidence interval 0.58–0.96) for a 25% recurrence risk reduction with the addition of neoadjuvant T-VEC (Figure 1) [55]. Side effects included flu-like illness and pyrexia and were consistent with the known T-VEC safety profile established by the OPTiM trial [54,55]. The application of T-VEC monotherapy as a neoadjuvant therapy in surgically resectable, high-risk melanoma is also being evaluated in a phase 2 trial, NCT04427306 (Table 1).

Theoretically, oncolytic viral therapies and ICIs should have synergistic effects since the viral infection increases susceptibility to immune surveillance and facilitates antigen-presenting cell (APC) processing [49,56]. A phase 2 trial in which CD8+ T cell levels were monitored in response to intralesional T-VEC demonstrated that, while CD8+ T cell levels were not associated with response, a 2.4-fold increase in CD8+ T cells was observed in non-injected lesions and the CD8+ T cells identified in non-injected lesions had increased expression of PD-1, programmed death-ligand 1 (PD-L1), and cytotoxic T-lymphocyte-associated protein 4 (CTLA-4) checkpoints [57]. In a 2018 phase 2 trial (NCT01740297) of 98 patients who received intratumoral T-VEC with intravenous ipilimumab versus 100 who received systemic ipilimumab alone, ORR was significantly improved with the combination, 39% vs. 18%, respectively, without a significant increase in toxicity, the most common AEs in the combination arm being fatigue (59%), chills (53%), and diarrhea (42%) (Figure 1) [58]. However, in 2022, a phase 3 (NCT02263508) randomized controlled, double-blinded trial evaluating intralesional T-VEC with systemic pembrolizumab versus an intralesional placebo with systemic pembrolizumab in 692 patients with Stage IIIB-IVM1c unresectable, anti-PD-1-naïve melanoma found no significant difference in overall survival (OS) or progression-free survival (PFS) and similar rates of grade 3 or greater treatment-related AEs between the two groups [59]. However, it is important to note that the lack of statistical significance in survival may in part be due to the inclusion of stage IVM1b and IVM1c patients in this trial when the most benefit from T-VEC has been demonstrated in stage IIIB-IVM1a in the past [60].

An ongoing phase 2 trial (NCT04068181) is evaluating the combination of intralesional T-VEC with systemic pembrolizumab in patients who have previously progressed on anti-PD-1 therapy (Table 1). Preliminary results have shown limited efficacy in those who previously progressed on anti-PD-1 in a locally recurrent or metastatic setting, 0% ORR in Cohort 1 (primary resistance, *n* = 26) and 7% ORR in Cohort 2 (acquired resistance, *n* = 15), but more promising results in those who progressed after receiving anti-PD-1 only in the adjuvant setting, 40% ORR in Cohort 3 (disease free < 6 months, *n* = 15) and 47% ORR in Cohort 4 (disease free ≥ 6 months, *n* = 15) [25]. The most common treatment-related AEs in all cohorts combined were pyrexia (30%), fatigue (16%), and flu-like illness (16%) with 13% experiencing grade 3 or higher treatment-related AEs. 

The combination of intratumoral T-VEC with systemic ICI is also being examined in the neoadjuvant setting. A phase 2 trial, NCT04330430, is evaluating neoadjuvant T-VEC with nivolumab for resectable Stage IIIB-IVM1a melanoma (Table 1). In patients with clinically node-positive disease, neoadjuvant intranodal T-VEC injection with pembrolizumab is being evaluated in a phase 2 trial, NCT03842943.

The addition of T-VEC to other treatment modalities also provides an avenue for therapeutic benefit. Intralesional T-VEC with and without radiotherapy is under investigation in a phase 2 trial in Stage IIIB or higher melanoma and other solid malignancies (NCT02819843) (Table 1). Additionally, a phase 1/2 trial (NCT03555032) examining the efficacy of intralesional T-VEC with isolated limb perfusion in stage IIIB-IVM1b melanoma as well as sarcoma is ongoing.

#### 2.4.2. RP1 (Vusolimogene Oderparepvec)

RP1 is a genetically modified HSV1 encoding GM-CSF and the gibbon ape leukemia virus fusogenic membrane glycoprotein with R sequence deletion (GALV-GP R-) that has been demonstrated to have potent antitumor activity in vitro, abscopal effects in murine models, as well as potentiated effects when administered with anti-mouse PD-1 antibody in murine models [61]. In humans, a phase 1/2 trial (NCT03767348) of intratumoral RP1 alone and in combination with intravenous nivolumab in advanced solid tumors, including Stage IIIB-IVM1c melanoma, is underway (Table 1). Interim results in cutaneous melanoma reported an ORR of 63% in eight anti-PD-1-naïve patients who received combination therapy and 38% in 16 patients who previously failed anti-PD-1 therapy [26]. More recent interim results of 75 patients who failed anti-PD-1/L1 therapy reported an ORR of 36% and DCR of 53% at a median follow-up of 10 months [27]. An additional phase 1b/2 trial of RP1 (NCT04349436) is ongoing for cutaneous malignancies, including locally advanced or metastatic melanoma, in solid organ transplant patients.

#### 2.4.3. OrienX010

OrienX010 is an oncolytic HSV engineered to express GM-CSF like T-VEC does; however, it uses a strain isolated in China, HSV1 CL1 [62]. It is undergoing phase 2 investigation (NCT04200040) as a single, intratumoral agent compared to systemic dacarbazine in treatment-naïve, unresectable Stage IIIB-IVM1b melanoma in China (Table 1).

#### 2.4.4. ONCOS-102 (Ad5/3-Δ24-GM-CSF)

Oncos-102 is an adenovirus that is genetically modified to express GM-CSF that has preferential tumor cell binding through desmoglein 2 receptors [63]. As with T-VEC, the intratumoral GM-CSF increases the antitumor response by increasing NK and CD8+ T cells. Furthermore, combination with pembrolizumab in murine models has increased antitumor activity, thought to be related to increased immune surveillance against tumor-antigens exposed by viral cell lysis. A phase 2 trial (NCT05561491) of intratumoral ONCOS-102 alone or in combination with intravenous anti-PD-1 inhibitor, balstilimab, in anti-PD-1-refractory unresectable or metastatic melanoma is not yet accruing (Table 1).

#### 2.4.5. BT-001

BT-001 is an oncolytic vaccinia virus containing genes encoding human GM-CSF and the human recombinant anti-CTLA-4 antibody [64]. This allows for T-VEC-like effects of local GM-CSF delivery as well as local immune checkpoint blockade from delivery of the anti-CTLA-4 antibody. In murine models, it has also been demonstrated to have abscopal effects despite low systemic levels of the anti-CTLA4 antibody. An ongoing, phase 1/2a, multipart clinical trial, NCT04725331, is evaluating intratumoral BT-001 alone and in combination with systemic pembrolizumab in multiple solid tumors, including locally advanced or metastatic melanoma (Table 1).

#### 2.4.6. Canerpaturev (C-REV, TBI-1401, HF10)

C-REV is a strain of HSV1, HF10, with naturally occurring genomic alterations that result in preferential infection of and replication in tumor cells [65]. This causes cytolysis and intratumoral penetration of CD4+, CD8+, and NK cells. As previously discussed, oncolytic viruses are theorized to synergize with ICIs by increasing immune surveillance [49,56]. Therefore, intratumoral C-REV in combination with intravenous ipilimumab was investigated in a phase 2 trial (NCT02272855) in 44 evaluable patients with ipilimumab-naïve, stage IIIB-IV melanoma (Figure 1). The results presented in 2018 included an ORR of 41% and DCR of 68% [66]. Grade 3 or higher AEs were reported in 37%, though only 7% were attributed to C-REV and were classified as gastrointestinal, musculoskeletal, metabolism/nutrition, and vascular disorders. In 2019, the results of a similar phase 2 trial (NCT03153085) in 27 evaluable Japanese patients who had failed prior therapies were presented and included an 11% ORR and 56% DCR [67]. Unspecified severe AEs were seen in 36% of patients. Further investigation of the combination of intralesional C-REV with systemic ICI was undertaken in a phase 2 trial (NCT03259425) examining C-REV in combination with nivolumab in the neoadjuvant setting for resectable stage IIIB-IVM1a melanoma, but it was terminated in 2022 at the recommendation of the Data Safety Monitoring Committee. 

#### 2.4.7. Coxsackievirus A21 (CVA21, CAVATAK, V937, Gebasaxturev)

CVA21 is an enterovirus that preferentially infects melanoma cells since it binds to receptors that are overexpressed on melanoma cells, decay-accelerating factor (DAF) and intracellular adhesion molecule-1 (ICAM-1) [68]. A phase 2 trial (NCT01227551) of intratumoral CVA21 in 57 patients with stage IIIC-IV melanoma published in 2021 reported an ORR of 39% (unconfirmed per irRECIST) and 28% (confirmed) as well as 12-month PFS and OS of 33% and 75%, respectively (Figure 1) [69]. All treatment-related AEs reported were grade 1 or 2 with the most common being injection site pain, fatigue, and chills. Patients with stable or responding disease could continue on NCT01636882, the extension study.

The synergistic potential of CVA21 delivered both intratumorally and intravenously in combination with systemic pembrolizumab versus pembrolizumab alone in anti-PD-1-naïve stage III-IV melanoma is under phase 2 investigation (NCT04152863) (Table 1). The same combination is under investigation in the neoadjuvant setting in one arm of a multi-arm, phase 1/2 study (NCT04303169) comparing systemic pembrolizumab in combination with five different investigational agents in stage III melanoma.

#### 2.4.8. PVSRIPO (Lerapolturev)

PVSRIPO is a recombinant, live-attenuated poliovirus Sabin type 1 with human reovirus type 2 in the internal ribosomal entry site [70]. It preferentially enters melanoma cells, which have a high rate of CD155 poliovirus receptor expression, and stimulates the host antiviral immune response against the tumor cells. It is currently under investigation in a phase 2 randomized clinical trial (NCT04577807) as a single, intralesional agent or in combination with systemic anti-PD-1 therapy in unresectable, anti-PD-1/L1 refractory stage III-IVM1b (Table 1).

#### 2.4.9. OBP-301 (Telomelysin)

OBP-301 is a telomerase-specific replication-competent oncolytic adenovirus 5 in which human telomerase reverse transcriptase enables selective replication in tumor cells [71]. Preclinically, it has been shown to increase local infiltration of CD8+ T cells and APCs and decrease regulatory T cells in both injected and uninjected tumor sites. A phase 2a trial, NCT03190824, examining this drug as an intralesional therapy in unresectable Stage III-IV melanoma is ongoing (Table 1).

#### 2.4.10. Voyager-V1 (VV1, VSV-IFNβ-NIS)

VV1 is a vesicular stomatitis virus engineered to express interferon-beta (IFN-β), potentiating the immune response to viral cell lysis, as well as a sodium/iodide symporter (NIS) gene for tracking through single-photon emission computerized tomography (SPECT) or positron emission tomography (PET) imaging [72,73]. It has been shown to increase inflammation and T cell infiltration in injected and non-injected lesions in phase 1 trials, and clinical benefit was observed as a monotherapy and in combination with systemic ICI in ICI-refractory disease. A phase 2 trial (NCT04291105) is underway in anti-PD-1/L1-refractory solid tumors, including advanced or metastatic melanoma; one study arm includes intratumoral and intravenous VV1 with intravenous cemiplimab in melanoma (Table 1).

#### 2.4.11. Ad-p53

Ad-p53 is a recombinant human adenovirus with wild-type *p53*. It functions as a gene therapy, delivering the wild-type *p53* gene to tumor cells, which otherwise often suppress p53 function as a part of regulatory evasion [74]. It has been applied to other malignancies without any reported serious adverse events, the most common AEs being transient fever, flu-like symptoms, muscle aches, and injection site pain, and is approved in China for the treatment of head and neck squamous cell carcinoma. It is now under phase 2 investigation (NCT03544723) in combination with ICI in recurrent or metastatic solid tumors including melanoma (Table 1).

#### 2.4.12. RheoSwitch Therapeutic System

RheoSwitch is comprised of a replication-incompetent adenoviral vector (Ad-RTS-hIL-12, IXN-2001) administered via intratumoral injection in combination with an oral activator ligand (veledimex, IXN-1001) [75]. Preclinical models demonstrated increased intratumoral IL-12 and CD8+ T cells with a corresponding dose-dependent decrease in tumor volume. Preliminary results of a phase 1/2 trial (NCT01397708) of this therapeutic combination in unresectable Stage III–IV melanoma reported a similar increase in tumor IL-12 mRNA and tumor-infiltrating lymphocytes and well-tolerated dose escalation (Table 1) [28].

### 2.5. Toll-like Receptor 9 (TLR9) Agonists

TLR 9 agonists act to induce DC maturation; cytokine secretion; APC uptake, processing and presentation; NK cell activation; and T cell response, thereby improving tumor immunogenicity [76,77,78]. Additionally, this improves tumor cell susceptibility to ICIs as the antitumor response is dependent upon tumor immunogenicity [65,79,80,81,82].

#### 2.5.1. Tilsotolimod (IMO-2125)

TLR9 agonist, tilsotolimod, activates the Th-1-type immune response, which increases expression of ICIs by promoting the maturation of local APCs [83]. The results from 49 evaluable patients with Stage IIIC-IV, anti-PD-1 refractory melanoma treated with tilsotolimod and ipilimumab in a phase 2 trial (NCT02644967) published in 2020 included an ORR of 22% and median OS of 21 months (Figure 1) [84]. The most common AEs seen related to this combination were fatigue, nausea, and anemia with 48% experiencing grade 3 or 4 treatment-emergent AEs and 32% experiencing at least one serious AE. This combination was compared to systemic ipilimumab alone in anti-PD-1-refractory, stage III-IVM1c melanoma in a phase 3, randomized trial (NCT03445533); however, it was terminated in 2022 for a lack of improvement in the ORR and OS.

#### 2.5.2. SD-101

SD-101, a TLR9 agonist, is a synthetic cytosine–phosphate–guanine (CpG) oligonucleotide that exerts its effects by inducing the production of CD8+ T cells and T cell infiltration [85]. A phase 1/2 trial (NCT02521870) of SD-101 in combination with pembrolizumab in metastatic melanoma or head and neck squamous cell carcinoma was terminated by the sponsor in 2021 with no plans for additional investment in SD-101 development. 

#### 2.5.3. Vidutolimod (CMP-001)

Vidutolimod is comprised of CpG-A DNA, a TLR9 agonist, packaged in a virus-like particle, which ultimately induces IFN-α production by DCs and creates a Th-1-type immune response [86,87]. The combination of intralesional vidutolimod with systemic nivolumab is under phase 2 investigation (NCT04698187) in anti-PD-1-refractory and phase 2/3 investigation (NCT04695977) compared to systemic nivolumab alone in treatment-naïve, unresectable or metastatic melanoma (Table 1). 

The role of vidutolimod in the neoadjuvant setting is also being explored. Neoadjuvant intralesional vidutolimod in combination with intravenous nivolumab was evaluated in a phase 2 clinical trial (NCT03618641) in 30 patients with Stage IIIB-D, clinically node-positive melanoma (Figure 1) [88]. On surgical pathology, CR was seen in 50% with 70% having a >50% reduction in tumor volume. There were no dose-limiting toxicities, three patients experienced an unspecified grade 3 or 4 immune-related AE and two discontinued vidutolimod as a result. A similar trial of neoadjuvant intratumoral vidutolimod with systemic nivolumab versus nivolumab alone is being evaluated in an ongoing phase 2 study, NCT04401995, in stage IIIB-D melanoma, but with palpable nodal disease or nodal recurrence (Table 1). Neoadjuvant intralesional vidutolimod is also being investigated in combination with systemic pembrolizumab versus pembrolizumab alone in stage III, N1b-N3c, resectable melanoma in a phase 2 trial (NCT04708418).

#### 2.5.4. Cavrotolimod (AST-008)

Cavrotolimod (AST-008) is a spherical nucleic acid configuration, with densely and radially arranged oligonucleotides on liposomal nanoparticles, of a TLR9 agonist that has been shown, preclinically, to elicit a Th-1-type immune response similar to other TLR9 agonists [89]. Clinical investigation of intratumoral cavrotolimod in combination with systemic pembrolizumab is ongoing in a phase 1b/2 trial (NCT03684785) in solid tumors including anti-PD-1/L1-refractory, locally advanced or metastatic melanoma (Table 1). Interim phase 1b data reported that in 19 evaluable patients, the ORR was 21% and two of the four responders had anti-PD-1 refractory melanoma [29]. There were no dose-limiting toxicities and the most common treatment-related AEs were grade 1 or 2 injection site reactions. 

### 2.6. Toll-like Receptor 7/8 (TLR 7/8) Agonists

#### NKTR-262

NKTR-262, a TLR7/8 agonist, has similar effects to TLR9 agonists; it increases CD8+ T cell and NK cells, resulting in amplified tumor antigen release and presentation [90,91]. In combination with bempegaldesleukin, it has been shown to increase innate immune signaling and antigen presentation [92]. Therefore, the combination of intralesional NKTR-262 with systemic bempegaldesleukin with or without systemic nivolumab was under investigation in unresectable or metastatic solid tumors including melanoma in a phase 1/2 trial (NCT03435640); however, it was recently terminated by the sponsor based on phase 1 findings. 

### 2.7. Immune Checkpoint Inhibitors

Ipilimumab is an anti-CTLA-4 antibody that acts to disinhibit the antitumor T cell response and is FDA-approved for metastatic melanoma [93]. However, its efficacy as an intravenous therapy is often limited by significant systemic adverse effects [93,94]. Intratumoral delivery offers the potential to avoid systemic dosing and, in early clinical investigation, was well tolerated and elicited responses in combination with intratumoral IL-2 [95]. Specifically, of the 12 patients, none experienced dose-limiting toxicities, there were no grade 4 or 5 treatment-related AEs, and the most common were injection site reaction, pain, and ulceration as well as fatigue and chills. Intratumoral ipilimumab is now under investigation in combination with systemic nivolumab compared to intravenous administration of both drugs in a phase 1/2 trial (NCT02857569) in stage III-IV melanoma (Table 1). This concept is being expanded to include localized delivery of various ICIs in a phase 2/3 trial (NCT03755739) assessing the efficacy of intralesional and trans-arterial delivery of investigator’s choice ICIs compared to standard venous administration in advanced sold tumors, including melanoma. 

### 2.8. Other Therapeutics

#### 2.8.1. LL37

LL37 is an endogenous antimicrobial peptide that can increase DC and B cell recognition and binding of CpG oligonucleotides [96]. A phase 1/2 trial (NCT02225366) investigating intratumoral LL37 in Stage IIIB-IVA melanoma is awaiting publication (Table 1).

#### 2.8.2. Hiltonol (Polyinosinic-Polycytidylic Acid-Poly-I-Lysine Carboxymethylcellulose, Poly-ICLC)

Hiltonol is a toll-like receptor 3 and melanoma differentiation-associated protein 5 ligand, a synthetic double-stranded RNA mimic of a pathogen-associated molecular pattern [97]. In early clinical investigation with intratumoral and intramuscular injection, it was shown to increase intratumoral levels of CD4+ and CD8+ T cells, PD-1, and PD-L1. A phase 2 trial (NCT02423863) of intratumoral and intramuscular hiltonol alone or in combination with anti-PD-1/L1 in advanced, unresectable solid tumors, including melanoma, is now underway (Table 1). 

#### 2.8.3. APX005M

APX005M is a monoclonal antibody agonist of CD40. CD40 is present on APCs and macrophages and the CD40 ligand is on T cells; therefore, APX005M has the potential to activate CD8+ T cells and increase major histocompatibility complex (MHC) class I expression on tumor cells [98]. It is now under investigation as an intralesional agent in a phase 1/2 trial (NCT02706353) in combination with systemic pembrolizumab in ICI-naïve, unresectable stage III-IVM1c melanoma (Table 1).

#### 2.8.4. Dendritic Cell Therapy

DCs are APCs, which, in their mature state, express MHC-antigen complexes and costimulatory molecules, and elicit a T cell response [99]. Since DC activation is an important part of the antitumor immune response, autologous DCs have been a therapeutic avenue under investigation for many malignancies. A phase 1b/2 trial (NCT03325101) examining the efficacy of intratumoral autologous mature DCs in combination with cryosurgery and systemic pembrolizumab for anti-PD-1/L1-refractory unresectable stage III-IV melanoma is ongoing (Table 1). 

#### 2.8.5. INT230-6

INT230-6 is a combination of cisplatin and vinblastine with an amphiphilic penetration enhancer for targeted tumor cell delivery [30]. It has been shown to result in CD4+ and CD8+ T cell activation and DC recruitment with abscopal effects in preclinical models as well as synergistic effects when combined with ICIs. It is under phase 1/2 investigation (NCT03058289) in solid tumors, including treatment-refractory metastatic melanoma, alone and in combination with ICIs, and interim results show it is well-tolerated without dose-limiting toxicity or grade 4 or 5 treatment-emergent adverse events (Table 1) [30]. The most common treatment-related AEs reported were localized pain, nausea, fatigue, and vomiting. 

#### 2.8.6. Polidocanol

Polidocanol, an intralesional sclerosant that has long been used in the treatment of varicose veins [100], was under investigation as an intralesional therapy in Stage IIIB-IV melanoma (NCT03754140); however, the phase 2 study was recently withdrawn due to poor accrual.

## 3. Conclusions

There remains significant room for advancement in the treatment of advanced melanoma. In patients with lesions amenable to injection, intralesional therapy provides a promising option. Though intralesional therapies for melanoma were under investigation nearly 50 years ago, the field has seen the most growth within the past decade. Following the FDA approval of T-VEC in 2015, the field has continued to grow and evolve. In the past 5 years, we have seen numerous therapies under investigation, with most modulating or activating the host immune response in some way. This has followed the trend of systemic immunotherapy for advanced melanoma. While systemic ICIs have revolutionized the treatment of advanced disease, response and durability is limited and there can be significant treatment-limiting immune-related adverse events. Local delivery of therapeutics that have the potential to modulate the immune response at the site(s) of disease provide promise as a way to enhance efficacy, circumvent resistance, and limit systemic adverse effects, as we have seen in some of the trials discussed above. 

Unfortunately, not every intralesional regimen investigated has been successful. Perhaps the most notable and surprising were the recent results of NCT02288897, the phase 3, randomized controlled trial of intralesional T-VEC with systemic pembrolizumab versus an intralesional placebo with systemic pembrolizumab in which no survival benefit was seen [59]. Since this study was conducted in anti-PD-1-naïve advanced melanoma, it highlights the importance of identifying the appropriate timing of therapeutic regimens. Additionally the inclusion of stage IVM1b-IVM1c melanoma despite past trials demonstrating greater efficacy in less advanced (stage III-IVM1a) disease [60], emphasizes the importance of patient selection for future trials. Further to this, the trial outcome suggests that optimal benefit of T-VEC in combination with ICI may either be in the neoadjuvant, or second-line or later setting, as is currently under investigation. 

Additionally, multiple trials evaluating intralesional toll-like receptor agonists in combination with systemic immunotherapies in metastatic melanoma (NCT03445533, NCT02521870, NCT03435640) have recently been terminated due to their lack of efficacy or concern from the sponsor. Other TLR-9 agonists, vidutolimod and cavrotolimod, remain in phase 2 trials with systemic ICIs and bear watching. 

Ultimately, the major limitation of intralesional therapy is administration. Injection should be performed by an experienced provider and drug delivery is dependent on provider technique. T-VEC and other intralesional therapies have attempted to account for variability by providing specific dosing instructions based on lesion size and number; however, the nature of intralesional therapy makes it impossible to completely standardize dosing. For many patients, a significant barrier to receiving intralesional therapies is access to healthcare facilities in which they are offered. This is further exacerbated by frequent dosing schedules. In general, it seems that the cytokine-based therapies (IL-2) and TLR-9 agonists have the most intensive dosing regimens, while other therapies have less frequent dosing, but even the less intensive schedules are every 3–4 weeks.

Lastly, the intratumoral delivery of these therapies is advantageous in generating a response to specific disease sites and limiting systemic side effects; however, it also appears to limit efficacy in uninjected sites of disease. The abscopal effects of intralesional therapies are of great interest since in many cases of advanced disease, it is not possible to inject every tumor and microscopic disease may not be identified. While intralesional therapies have demonstrated a response in some sites of uninjected disease, to our knowledge, no studies of intratumoral agents alone have demonstrated an equivocal response in injected and uninjected lesions.

### Future Directions

In discussing the limitations and shortcomings of current intralesional therapies, we highlight areas for improvement and future directions. One of the greatest areas for advancement is sorting out the appropriate drug regimens and timing. There are numerous possible combinations of intratumoral and systemic therapies, even if only using the drugs that have been discussed here. We have already seen instances of regimens that appear promising in other settings or earlier phase trials that have not delivered expected results. Therefore, we should expect more clarity regarding effective timing and combinations in the coming years. 

Accessibility of intratumoral therapy is another avenue for future improvement. As more intratumoral therapies and combination regimens become FDA-approved or widely available, we can expect to see more providers administering intralesional therapies. Additionally, once more efficacious intratumoral therapies and combination regimens have been identified, accessibility can be further improved with optimization of the dosing schedules to make them more patient friendly where possible. 

Finally, the application of precision medicine to intratumoral therapy is largely unexplored. The use of autologous mature DCs as intralesional therapy is one of the few personalized therapies currently under investigation. With more widespread tumor genome sequencing, identification of targetable mutations, and development of targeted therapies, we can hope for a more individualized approach to intralesional therapy in the future. 

## Figures and Tables

**Figure 1 cancers-15-01404-f001:**
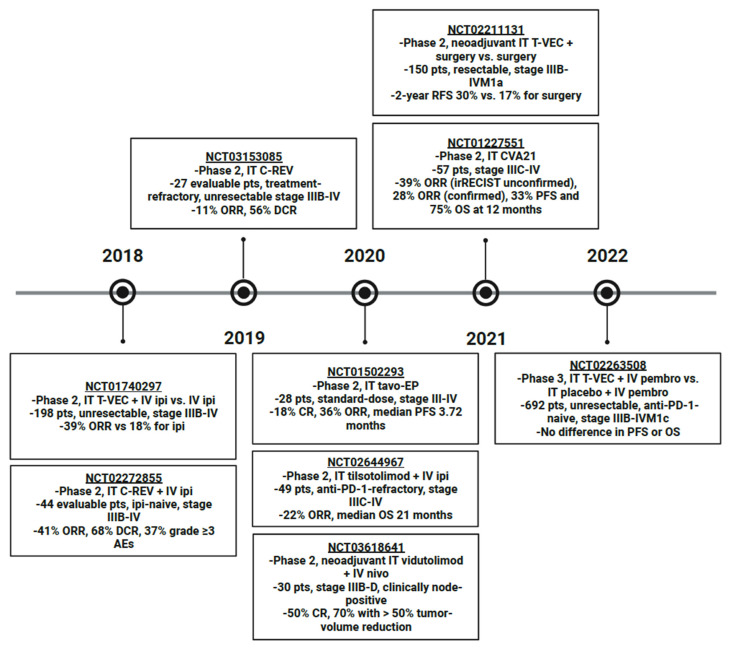
Timeline of Phase 2 or later clinical trials of intralesional therapy for advanced melanoma published in the last 5 years. Abbreviations: IT, intratumoral; IV, intravenous; T-VEC, talimogene laherparepvec; ipi, ipilimumab; pts, patients; ORR, overall response rate; C-REV, canerpaturev; DCR, disease control rate; AE, adverse event; tavo-EP, tavokinogene telseplasmid electroporation; CR, complete response; PFS, progression-free survival; PD-1, programmed cell death protein 1; OS, overall survival; nivo, nivolumab; RFS, recurrence-free survival; CVA21, coxsackievirus A21; irRECIST, immune-related response evaluation criteria in solid tumors; pembro, pembrolizumab.

**Table 1 cancers-15-01404-t001:** Ongoing Phase 2 or Later Clinical Trials of Intralesional Therapy for Locoregionally Advanced and Metastatic Melanoma.

Clinical Trial Number	Phase	Regimen	Melanoma Population	Intralesional Dosing Schedule	Preliminary Results
NCT02557321	1b/2	IT PV-10 + IV pembro vs. IV pembro alone	ICI-Naïve and ICI-refractory, unresectable, stage III–IV	Q3W x 5C	- 21 ICI-naïve, 67% ORR [22] - 19 ICI-refractory, 26% ORR [23]
NCT03474497	1/2	IT IL-2 + IV ICI + hypofractionated XRT	PD-1/L1-refractory, stage IV	2x/W x 4C	-
NCT03928275	2/3	IT IL-2 vs. IT IL-2 + IT BCG	Stage IIIC-IVM1a	Q2W x 4C	-
NCT03132675	2	IT Tavo-EP + IV pembro	Anti-PD-1-refractory, unresectable, stage III–IV	Days 1, 5, 8, Q6W, up to 18C	- 54 evaluable, 30% ORR [24] - 5.4% grade 3 treatment-related AEs
NCT02076646	1/2	IT L19IL2 + IV DTIC vs. IV DTIC alone	Stage IV (M1a-b in phase 2)	Days 1, 8, 15 Q3W	-
NCT03567889	3	IT Daromun + surgery + IV ID adjuvant vs. surgery + IV ID adjuvant	Neoadjuvant, resectable stage IIIB-C	Q1W, up to 4C	-
NCT02938299	3	IT Daromun + surgery vs. surgery	Neoadjuvant, resectable stage IIIB-C	Q1W, up to 4C	-
NCT04427306	2	IT T-VEC + surgery	Neoadjuvant, resectable, high-risk	Not specified	-
NCT04068181	2	IT T-VEC + IV pembro	Anti-PD-1-refractory, unresectable, stage IIIB-IV	Q3W, up to 35C	- 26 primary resistance in recurrent/metastatic setting, 0% ORR [25] - 15 acquired resistance in recurrent/metastatic setting, 7% ORR - 15 resistance in adjuvant setting with <6 mos. disease free, 40% ORR - 15 resistance in adjuvant setting with ≥6 mos. disease free, 47% ORR
NCT04330430	2	IT T-VEC + IV nivo	Neoadjuvant, resectable stage IIIB-IVM1a	Q3W x 1C then Q2W x 3C	-
NCT03842943	2	IN T-VEC + IV pembro	Neoadjuvant, clinically node-positive, resectable, stage III	Q3W up to 6 mos.	-
NCT02819843	2	IT T-VEC ± XRT	Unresectable, stage IIIB-IV	Q3W x 1C then Q2W x 7C	-
NCT03555032	1/2	IT T-VEC + ILP	Amenable to ILP, stage IIIB-IVM1b	Q2-3 Weeks x 3 doses	-
NCT03767348	1/2	IT RP1 ± IV nivo	Stage IIIB-IVM1c	Not specified	- 8 anti-PD-1-naïve, 63% ORR. [26] - 16 anti-PD-1-refractory, 38% ORR - 75 anti-PD-1/L1-refractory, 36% ORR, 53% DCR [27]
NCT04349436	1b/2	IT RP1	Treatment-refractory, locally advanced or metastatic, solid organ transplant recipients	Q2W	-
NCT04200040	2	IT OrienX010 vs. IV DTIC	Treatment-naïve, unresectable, stage IIIB-IVM1b	Q2W	-
NCT05561491	2	IT Oncos-102 ± IV balstilimab	Anti-PD-1-refractory, unresectable or metastatic	Not specified	-
NCT04725331	1/2a	IT BT-001 ± IV pembro	Locally advanced or metastatic	Not specified	-
NCT04152863	2	IT & IV CVA21 + IV pembro vs. IV pembro	ICI-naïve, stage III-IV	Day 1, 3, 5, 8 Q4W x 1C, then Q3W x up to 7C	-
NCT04303169	1/2	IT CVA21 + IV pembro + surgery vs. other investigational drugs + IV pembro + surgery	Neoadjuvant, treatment-naïve, resectable, stage IIIB-D	Not specified	-
NCT04577807	2	IT PVSRIPO ± IV anti-PD-1 therapy	Anti-PD-1/L1-refractory, unresectable, stage III-IVM1b	Q1W x 7C then Q3-4W	-
NCT03190824	2a	IT OBP-301	Unresectable, stage IIIB-IV	Q2W up to 13C	-
NCT04291105	2	IT and IV VV1 + IV cemiplimab ± ipi	Anti-PD-1/L1-refractory, advanced or metastatic	Q4W x 1C, then Q3W	-
NCT03544723	2	IT Ad-p53 + IV ID ICI	Recurrent or metastatic	Not specified	-
NCT01397708	1/2	IT RheoSwitch	Unresectable, stage III-IV	Q3W up to 6 C	- Dose-escalation well tolerated, most common AEs were chills, pyrexia [28]
NCT04698187	2	IT vidutolimod + IV nivo	Anti-PD-1-refractory, unresectable or metastatic	Q1W x 7C then Q3W	-
NCT04695977	2/3	IT vidutolimod + IV nivo vs. IV nivo	Treatment-naïve, unresectable or metastatic	Q1W x 7C then Q3W	-
NCT04401995	2	IT vidutolimod + IV nivo vs. IV nivo	Neoadjuvant, palpable nodal disease or nodal recurrence, stage IIIB-D	Q1W x 7C	-
NCT04708418	2	IT vidutolimod + IV pembro + surgery vs. IV pembro + surgery	Neoadjuvant, resectable, stage III N1b-N3c	Day 1, 8, 15 Q3W x 2C then Q3W x 4C	-
NCT03684785	1b/2	IT cavrotolimod + IV pembro	Anti-PD-1/L1-refractory, locally advanced or metastatic	Not specified	- 19 evaluable phase 1b, 21% ORR (2 of 4 responders had anti-PD-1-refractory melanoma) [29]
NCT02857569	1/2	IT ipi + IV nivo vs. IV ipi + IV nivo	Unresectable, stage III-IV	Q3W up to 4C	-
NCT03755739	2/3	IT ID ICIs vs. trans-arterial ID ICIs vs. IV ID ICIs	Advanced	Q3W	-
NCT02225366	1/2	LL37	Unresectable, stage IIIB-IVA	Q1W up to 8C	-
NCT02423863	2	IT hiltonol + IM hiltonol ± IV anti-PD-1/L1	Unresectable, advanced	IT Day 1, 3, 5 x 1W then IM 2x/W	-
NCT02706353	1/2	IT APX005M + IV pembro	ICI-naïve, unresectable, stage III-IVM1c	Q3W up to 4C	-
NCT03325101	1b/2	IT DCs + cryosurgery + IV pembro	Anti-PD-1/L1-refractory, unresectable, stage III-IV	Apheresis then Q3W x 2C	-
NCT03058289	1/2	IT INT230-6 ± IV ICI	Treatment-refractory, metastatic	Q4W x 5C	- No dose-limiting toxicity, grade 1–2 injection site pain most common [30]

Table 1 abbreviations: IT, intratumoral; IV, intravenous, pembro, pembrolizumab; ICI, immune checkpoint inhibitor; Q, every; W, week; x, times; C, cycle; ORR, overall response rate; IL-2, interleukin-2; XRT, external beam radiation therapy; PD-1, programmed cell death protein 1; PD-L1, programmed death-ligand 1; BCG, bacillus Calmette–Guerin; Tavo-EP, tavokinogene telseplasmid electroporation; AE, adverse event; DTIC, dacarbazine; ID, investigator’s discretion; T-VEC, talimogene laherparepvec; IN, intranodal, ILP, isolated limb perfusion; RP1, vusolimogene oderparepvec; CVA21, coxsackievirus A21; VV1, voyager V1; ipi, ipilimumab; Ad-p53, adenoviral p53; nivo, nivolumab; IM, intramuscular; DC, dendritic cell.

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
