# Peer review of "Advances in Intralesional Therapy for Locoregionally Advanced and Metastatic Melanoma: Five Years of Progress"

_cancers, 2023, doi:10.3390/cancers15051404_

Round 1

Reviewer 1 Report

This review by DePalo and Zager represents a comprehensive overview of this emerging area of research in advanced cutaneous melanoma. The manuscript is wellwritten and describes the major advances in intralesional/locoregional therapy the last years. Thus, a highly needed summary of the field. The authors are focused on describing the different agents used giving mechanism of action, preclinical findings and clinical efficacy which seems reasonable.

In order to potentially improve the manuscript I suggest the following:

1.       Please further describe how the selection of agents/trials included was done.

2.       Data on Safety and tolerability is very little described. This should be considered to be included where applicable and where data exist. Eg. for TVEC, IL-2 and TLR modulating agents there should be substantial data. Only local side-effects or also systemic side effects? What type? Immunereleted effects or other? Need for dose interuption or termination etc.  Manageable? Giving mild toxicity or grade 3-4 toxicity is not enough.

3.       Dose level and dose frequency is stated as one critical issue. What about route for local administration using eg. intrethecal, intrahepatic, intraportal, intraabdominal application? Are ther any initiatives in this direction that could be included in this review?

Author Response

We greatly appreciate the time and consideration you have put into your review and comments. Our responses are numbered to correspond with your comment numbers and are as follows:

  1. We have added additional information to the introduction section detailing the criteria used to identify and select the clinical trials discussed in this review.
  2. Where available, we have added additional safety and tolerability data. This includes additions to the following sections: PV-10, IL-2, Tavo, Daromun, T-VEC, RP1, C-rev, CVA21, VV1, ADp53, Rheoswitch, Tilsotolimod, Vidutolimod, Cavrotolimod, Ipi, INT230-6.
  3. While it is interesting to consider these alternative methods of drug delivery, those do not necessarily fit within the scope of intralesional therapy and would be more appropriate for a review of regional therapies.

Reviewer 2 Report

It is a comprehensive review about intralesional treatment for melanoma. It contains relative complete overview of all intralesional treatment. To really stand out I think this would be a topic for a systematic review with a systematic approach with clear-cut inclusion criteria which intralesional treatments are included and than it would be certain that it would be complete for the topic (with of course a selection made on inclusion and exclusion criteria). Now, it is not fully clear why these treatments were included. 

For TVEC, which I think is the most studied and already used in clinical practice I think they are a bit short on the mechanism of working. They focus on GM-CSF, while it also has oncolytic capacities and a mechanism to selectively go into melanoma cells and enhance the immune system by letting HSV be presented in the MHC molecule. All T-cell mediated mechanisms which they say later, but I think they can point this out more clear. 

Author Response

Thank you for the thoughtful suggestions. In response, we have added additional information to the introduction section detailing the criteria that were used to identify the clinical trials that we have discussed. While we agree that a formal systematic review would be more powerful, the early nature of the trials discussed (the majority, 37 trials, are still in process with no data or only preliminary data available) limits our ability to conduct a formal systematic review as defined by the PRISMA checklist.

We have also added additional information regarding the mechanism of action of TVEC in response to the second comment.